# Lipid Nanoparticles and Liposomes for Bone Diseases Treatment

**DOI:** 10.3390/biomedicines10123158

**Published:** 2022-12-07

**Authors:** Alexandra-Cristina Burdușel, Ecaterina Andronescu

**Affiliations:** 1Department of Science and Engineering of Oxide Materials and Nanomaterials, Faculty of Chemical Engineering and Biotechnologies, University Politehnica of Bucharest, 1–7 Gheorghe Polizu Street, 011061 Bucharest, Romania; 2Academy of Romanian Scientists, Splaiul Independentei 54, 050044 Bucharest, Romania

**Keywords:** lipid nanoparticles, bone disease, liposome, osteoporosis

## Abstract

Because of their outstanding biocompatibility, sufficient capacity to control drug release, and passive targeting capability, lipid nanoparticles are one of the world’s most widely utilized drug delivery systems. However, numerous disadvantages limit the use of lipid nanoparticles in clinical settings, especially in bone regeneration, such as challenges in transporting, storing, and maintaining drug concentration in the local area. Scaffolds are frequently employed as implants to provide mechanical support to the damaged area or as diagnostic and imaging tools. On the other hand, unmodified scaffolds have limited powers in fostering tissue regeneration and curing illnesses. Liposomes offer a solid foundation for the long-term development of various commercial solutions for the effective drug delivery-assisted treatment of medical conditions. As drug delivery vehicles in medicine, adjuvants in vaccination, signal enhancers/carriers in medical diagnostics and analytical biochemistry, solubilizers for various ingredients as well as support matrices for various ingredients, and penetration enhancers in cosmetics are just a few of the industrial applications for liposomes. This review introduces and discusses the use of lipid nanoparticles and liposomes and the application of lipid nanoparticles and liposome systems based on different active substances in bone diseases.

## 1. Introduction

Treating bone diseases such as osteoarthritis, osteoporosis, and bone cancer often results in reduced efficiency and/or adverse reactions because it is not specifically targeted to the site of action. Using an appropriate carrier should improve drug delivery to the site of bone disease [1,2].

The present challenge is figuring out how to enhance nanoformulations that are safe, effective, and scalable enough to be manufactured on a big scale and pushed to clinical utilization [3]. Nanoformulations are indisputably valuable tools for drug delivery [4]. Since lipid nanoparticles are frequently viewed as non-toxic, biocompatible, and simple to create formulations, they have gained popularity in this context [5]. Lipid nanocarriers are increasingly used in the pharmaceutical industry to transport and deliver a variety of therapeutic agents, from biotechnological products to small drug molecules [6].

Bone tissue has comparably inadequate blood flow, making it harder to treat diseases such as bone metastases. In advanced stages of many malignancies, including breast [7], prostate [8], and lung cancers [9], bone metastases are a prevalent consequence. It can cause bone pain, hypercalcemia, pathological fractures, and bone deformities, reducing the quality of life due to the increased activity of osteoclast cells. The most popular treatments for metastatic bone cancers today are surgery, radiation therapy, and chemotherapy, either separately or in combination. Chemotherapy has increased patient survival rates and quality of life, but because of its poor targeting ability, it must be administered frequently and at high doses, which can have serious side effects [10].

Liposomes [11], lipid nanoparticles [4], micelles [12], and exosomes [13] are examples of nanoscale lipid formulations. Regenerative hydrophilic and lipophilic drugs can be encapsulated in lipid nano-formulations to increase their bioavailability. Surface modification can be used to target drugs so they collect in a particular area increasing efficiency. As a result of recent research, a group of scientists has successfully repaired tissue injury utilizing lipid nano-formulations [14].

Taking into account the above, this review will present the use of lipid nanoparticles and liposomes in bone tissue engineering.

## 2. Lipid Nanoparticles

Early in the 1990s, lipid matrix nanoparticles (LNPs), which are employed as nanocarriers, were initially created [15]. Their synthesis represented a significant advance in the hunt for a new class of nanocarriers. They can be utilized to promote drug bioavailability, provide regulated release of active components, enhance intracellular permeability, and regulate drug delivery to target areas due to the many possibilities of their surface alterations [16].

Colloidal particles made up of a lipid matrix maintained by surfactants are lipid-based nanoparticles. They are known as lipid colloidal carriers [17] and range in size from 40 nm to 1000 nm. They are stable at room temperature. Surfactants, solid lipids, cosurfactants (if required), and active compounds are the typical components of LNP carriers. It is important to consider the lipids’ makeup. Fatty alcohols, fatty acid esters, acylglycerols, and combinations of acylglycerol esters are the most widely utilized lipids [18].

One significant class of nanocarriers used to transport medicinal and diagnostic compounds is called LNPs. The main advantages of LNPs over other nanocarriers classes are biocompatibility, biodegradability, safety, and the incorporation and delivery of both hydrophobic and hydrophilic molecules, the latter of which allow the theranostic concept and addresses the issue of real-time interaction between diagnosis and therapy [19].

This class of nanoparticles is represented by liposomes, niosomes, transfersomes, nanoemulsions [20], solid lipid nanoparticles (SLNs) [21], lipid nanocapsules (LNCs) [22], nanostructured lipid carriers (NLCs) [23], lipid-based micelles [24], core–shell lipid nanoparticles (CLNs) [25], and hybrid systems. Due to their resemblance to biological and natural components, these nanosystems are considered bioinspired and show tremendous promise as carriers for nanotherapeutic applications [26].

Lipids and surfactants, cholesterol, phospholipids, solid and liquid lipids (oils and fats), and non-ionic and other surfactants, are the major components of LNPs [27]. The components frequently self-assemble into the liquid crystalline phase in these situations, which offers advanced dynamics and metastable phases that are crucial to the systems’ performance [27]. LNPs may also comprise lipids and materials with unique physicochemical properties that result in particular self-assembly, features, and biological uses. Ionizable cationic lipids are the most recent and well-known example [28]. These lipids have a pKa < 6.5 and are non-charged in physiological pH, but they become charged during development. The full potential of this group of lipids has been utilized in the creation of some recent COVID-19 vaccines. They create complexes with interior compartments that are hydrophobic and hydrophilic when combined with genetic material, such as siRNA or mRNA. The hydrophilic compartments carry the genetic material, but their form changes depending on the lipid combination used (which may also include a non-ionizable “helper” lipid), the N/P ratio (lipid amine to nucleotide phosphate), and the mixing procedure [29]. Nano-adjuvants, RNA-based vaccinations, and viral vectors are currently the technologies used in COVID-19 vaccines. For the complexation of the genetic material, the mRNA vaccines include patented ionizable cationic lipids, helper phospholipids, cholesterol, and a PEGylated surfactant or lipid that offers biological stability and stealth qualities [30]. The nano-adjuvant platform uses an immune-stimulating complex (ISCOM) matrix made of *Quillaja saponins*, cholesterol, and phospholipids as an adjuvant and a recombinant protein-based nanoparticle that carries the virus spike protein [31]. Clinical evidence supports the efficacy of the immune stimulatory complexes, or ISCOMs, as additional vaccine delivery vehicles with strong adjuvanticity. They self-assemble in solution at well-defined ratios of a protein antigen, cholesterol, and phospholipid and are around 40–60 nm in size. These substances to be employed for ISCOM synthesis include zwitterionic phospholipids, neutral or positively charged dimethylaminoethane-carbamoyl (DC)-cholesterol, and cationic dioleoyl-trimethyl ammonium-propane (DOTAP). These hollow-center, cage-like particles are also utilized to capture hydrophobic antigens [32].

As non-viral vectors, liposomes are also utilized in gene therapy. Due to the cationic surface’s ability to facilitate complex formation with the negatively charged DNA structure, only positively charged liposomes are currently employed for this purpose [33].

By employing liposomes as non-viral vectors, the unaltered gene can be delivered to the target cell. The transport of proteins and peptides into the cytoplasm of dendritic cells by endocytosis has extensively used liposome vesicles [5]. Since they do not contain any proteins, liposomes have an advantage over non-viral carriers and vectors in preventing immunological reactions. Their drawback, on the other hand, is that they might build up in a living body and fail to accomplish their purpose [34].

Bioactive lipids are a diverse class of substances known to act as inflammatory mediators [35]. They have a significant role in the pathogenesis of inflammatory illnesses, including lysophosphatidic acid (LPA), sphingosine 1-phosphate (S1P), and eicosanoids (prostanoids such as PGE2 and leukotrienes such as LTB4, LTC4, and LTD4) [36]. These bioactive lipids, however, have long been thought to play a role in cancer due to their capacity to affect the pro-inflammatory environment of tumors and their capability to act directly on tumor cells, promoting cell proliferation, migration, and survival. LPA, which has a glycerophosphate backbone coupled to a single fatty acid chain, is the most fundamental phospholipid. LPA is generated at the bone location, as we have previously shown in the case of bone metastases (Figure 1) [35,36].

LPA has an impact on both osteoclasts and cancer cells, hastening the progression of osteolytic lesions. LPA plays a crucial role in the development of bones. Although adipocytes and osteoblasts are two potential sources that could explain bone homeostasis and metastasis, the origin of LPA in bone is yet unknown [37].

The most prevalent primary malignant bone tumor in children is osteosarcoma (OS). Current therapies cannot stop disease progression in patients with metastasis, and their survival rate remains between 10% and 30% [38]. In this context, it has been suggested that using nanomedicine, particularly lipid nanoparticles (LNPs), could help cure many diseases, including bone cancer, without being constrained by current medical practices [39]. Nanocarriers have the benefit of being administered orally as opposed to intravenously, as in the case of LNPs, making them useful as therapeutic vehicles for anti-neoplastic drugs. Due to the enhanced permeability and retention (EPR) effect or active targeting by surface modifications, these nanoscaled drugs can be specifically targeted towards the desired site of action once in the systemic circulation, achieving a sustained release of the drug in the affected area and increasing their bioavailability inside the tumor [40]. Systemic toxicity should be decreased due to the use of LNPs, necessitating fewer dosages of anti-neoplastic drugs to treat the illness. To increase the effectiveness of chemotherapy in both primary and metastatic human osteosarcoma cells, Gonzales-Fernandez et al. suggested a drug delivery system based on lipid nanoparticles and edelfosine (ET), a novel active agent capable of specifically targeting cancer cells. Based on their greater absorption in primary and metastatic OS cells, the study discovered that encapsulating ET in LNPs considerably increased the drug’s efficacy against OS cells [41].

When bones are formed naturally, CaP nanoparticles help to mineralize the bone matrix. In addition to producing collagen and other bone proteins, mature osteoblasts (natural bone-building cells) also promote the mineralization of collagenous matrix by making membrane-bound vesicles called matrix vesicles (MVs) [42]. Using an emulsification technique, Chaiin et al. introduced a novel method for creating lipid nanoparticles (LNPs), which can imitate the MV by self-calcifying bodily fluids. Researchers also examined the cholecalciferol (vitamin D3)-loaded NPs’ encapsulation and release profiles. The percentages of encapsulation and release profiles of NPs loaded with cholecalciferol (vitamin D3) were also examined. The LNPs merely bind to collagen scaffolds, stimulating the gradual deposition of CaP and the release of lipophilic compounds. Due to their adaptability and low toxicity, LNPs are perfect for use in bone tissue engineering scaffolds [43,44].

Critical-sized bone defects are frequently left behind by traumatic fractures, tumor excision, and congenital deformities and are not likely to heal independently. To create in vitro bone replacements and get beyond the drawbacks of natural bone grafts, bone tissue engineering is an attractive technique [45]. In a groundbreaking study, a new bone engineering scaffold based on gelatin methacrylate (GelMA) hydrogel was constructed using a two-step process: Resveratrol, a substance that can enhance osteogenic differentiation and bone formation, was first incorporated into solid lipid nanoparticles (SLNs) (Res) [45]. To create the final Res-SLNs/GelMA scaffolds, these particles were subsequently encapsulated in GelMA at several concentrations (0.01%, 0.02%, 0.04%, and 0.08%). The impact of these scaffolds on bone healing in rats with cranial deficiencies and osteogenic differentiation of bone marrow mesenchymal stem cells [21]. Using emulsification and low-temperature solidification methods, it was successfully created Res-SLNs. Res-SLNs/GelMA hydrogels, which serve as scaffolds for bone tissue engineering, were subsequently made using the Res-SLNs. The scaffolds’ topologies allowed for slow, continuous release of Res, which improved osteogenic differentiation in vitro and bone regeneration in vivo. Additionally, eight weeks following surgery, rat cerebral deficiency sites transplanted with Res-SLNs/GelMA hydrogel scaffolds entirely recovered. For critical-sized bone defects, this study offers a possible therapy alternative [21].

Recently, among chemotherapeutics, vanadium compounds have become non-platinum antitumor agents [46]. In this regard, one of the most promising vanadium anticancer complexes was found to be Metvan ([V^IV^O(Me_2_phen)_2_(SO_4_)]) [38]. In this study, the biopharmaceutical profile of the metvan material in terms of bioavailability, degradation, solubility, and cell uptake was optimized by encapsulating it in well-designed and produced nanostructured lipid carriers (NLCs). The best nanoparticle composition for Metvan delivery was found using a quality-by-design methodology. In vitro, metvan-loaded NLCs outperformed metvan alone in inhibiting osteosarcoma cancer cell lines. Compared to free Metvan therapy, NLCs-Mv demonstrated the most significant cytotoxic effects, dependent on drug loading. We postulated that a rise in cellular effects is associated with securing the metal complex inside the nanoparticle core while the medication is constantly administered. Our study raises the question of whether using cutting-edge drug delivery techniques could lessen Metvan’s side effects in cancer treatment [47,48].

The burst release of the payload before it reaches the target site is the most challenging part of creating a drug transporter [49]. These lipid-based delivery methods have several benefits, but the biggest problem is keeping the medications intact while loading them effectively without losing any of them [50]. The entrapped drug molecule can remain stable for up to 6 months with the proper drug carrier and composition, but if it is in solution form, it can degrade in 30 days [51]. By blocking the mevalonate pathway, non-hydrolyzable pyrophosphate analogs known as bisphosphonates prevent cells from growing and signaling. Alendronate sodium is a bisphosphonate that contains nitrogen (second-generation bisphosphonate). As a result of its strong affinity for bones, it may help to slow down bone turnover. Therefore, it can be used as a targeted moiety in treatment plans for bone metastases [52]. Solid lipid nanoparticles, in addition to liposomes, are adaptable enough to deliver bisphosphonates. According to one investigation, the active molecule’s transportation was significantly enhanced when bisphosphonate was coupled with the preferred surfactant. It is well known that the surfactant plays the largest role in lipid nanoparticle composition [22].

## 3. Liposomes

Due to their flexibility and biocompatibility with both hydrophobic and hydrophilic and c drug loading, liposomes have been extensively employed as drug carriers [53]. The gold standard technique for avoiding rapid absorption and accumulation by phagocytic cells of the reticulo-endothelial system is surface modification with a hydrophilic polymer, such as polyethylene glycol (known as PEGylation, for example, liposomal doxorubicin, Doxil) [54]. Tumor environmental stimuli-triggered drug release has gained favor recently as a way to boost cancer therapy effectiveness. For instance, encapsulated drugs have been burst released using redox-sensitive nanoparticles with disulfide bonds (-SS-) in response to the intracellular redox potential of malignancies [55].

Liposomes are spherical vesicles that range in size from 0.025 µm to 2.5 µm. They are made up of one or more phospholipid bilayers made of natural phospholipids, cholesterol (giving the stability of liposomes), and are separated by spaces that are filled with water [56]. Phosphatidylcholine (PC), phosphatidylglycerol (PG), di-stearoyl phosphatidylcholine (DSPC), dipalmitoylphosphatidylcholine (DPPC), stearyl amine, diacetyl phosphate, cholesterol (CH), or mixes thereof are often the substances that make up the lipid bilayer [57].

A key factor in determining the half-life of liposomes in the bloodstream is the vesicle’s size. Drug encapsulation is significantly influenced by the liposome’s size and the number of its bilayers [58].

Drug delivery is the principal application for liposomes. Using active ingredients enclosed in carriers and administering them at doses that guarantee delivery to the target areas without harming healthy tissues is one way to increase the efficacy of treatment [59].

Comparing liposomal drug carriers to conventional pharmacological compounds, there are many advantages. When the medicine is released into the body, they guard it against deterioration and early metabolization [60]. Encapsulation of active substances in liposomes can be obtained by two methods:passively—the active substance is encapsulated during liposome formation;actively—the active substance is encapsulated after the liposomes are formed [61].

For the treatment of fracture healing problems, Zhou et al. created a salvianic acid A (CAS#: 7682-21-4)-loaded bone targeting liposome formulation (SAA-BTL) using pyrophosphorylase cholesterol (cholesterol-PPi) as the targeting ligand [62]. Additionally, they revealed a brand-new bone-targeting liposome (BTL) that uses cholesterol-PPi as the targeting ligand. The SAA-BTL could bind to the bone via cholesterol-strong PPi’s chelation to bone apatite at various bone surfaces (including the growth plate, trabecular bone, and cortical bone), resulting in an important local distribution of high concentrations of SAA and enhanced retention (lasting longer than 20 days for one inject) in a glucocorticoid-induced delayed fracture model (HA) [10]. Through controlling HDAC3-mediated endochondral ossification, it has been demonstrated that SAA and SAA-BTL effectively stimulate osteogenesis and chondrogenesis during the healing of fractures and hasten the conversion of cartilage to bone [63].

Drug delivery methods frequently use liposomes because of their nontoxicity, biocompatibility, and biodegradability as a drug carrier. Due to its high biocompatibility and hydrophobicity, cholesterol is a key component in liposome production [64]. The capacity to regulate the fluidity of the phospholipid bilayer in the liposome, lower membrane permeability, stop drug leakage, prevent phospholipid oxidation, and increase membrane flexibility all help maintain the liposome’s stability [65]. Delivery of drugs targeting the bone may increase therapy effectiveness while simultaneously lowering drug dosage. To more efficiently transport PTX to bone metastases and improve the distribution of paclitaxel (PTX) in bone, a novel glutamic hexapeptide-folic acid (Glu6-FA) derivative was created and synthesized. In this study, we developed a Glu6-FA-modified liposomal drug delivery system. Thanks to glutamic oligopeptides, the liposome formulation showed bone affinity, and FA modification on the surface of the particles let the liposomes specifically recognize tumors [66]. The PTX-Glu6-FA-Lip possesses improved cytotoxicity, high HAP binding efficiency, and stability. More importantly, it was discovered that PTX-Glu6-FA-Lip increased the accumulation of loaded paclitaxel in metastatic bones during the in vivo metastatic bone targeting examination [67].

This study describes the successful encapsulation of the hydrophobic biomolecule curcumin into a liposome and the integration of the liposome within a 3D-printed TCP scaffold. The longer drug release from the liposome helps with increased bioavailability, therapeutic index, and toxicity reduction, while the scaffold offers mechanical support for cell adherence [68]. This research ushers in a new era of integration, in which cutting-edge 3DP technology is joined with the secure and effective use of alternative medicine, potentially producing a bone tissue manufacturing tool that is more efficient. Curcumin-encapsulated liposomes in 3D-printed bone tissue engineering scaffold reduce in vitro osteosarcoma cells by 96% compared to untreated samples. Additionally, it was non-cytotoxic to healthy osteoblast cells and promoted filopodial prosthesis growth on the scaffold surface by enhancing adhesion and proliferation [69].

Liposomes can deliver drugs directly to the area where they work and keep them there for a long time without causing harm [70]. The characteristics of liposomes can change by altering the lipid makeup. After receiving FDA approval, various liposome formulations for anticancer medications were effectively launched on the market [71]. Particles loaded with gentamycin and vancomycin and incorporated into liposomes are used to make scaffolds [56]. The development of stem cell osteoblasts might benefit from incorporating bioactive aspirin into a liposome delivery method [72]. The initial drug dosage, as well as the chemical and physical drug characteristics, are important contributors to the ineffectiveness of encapsulation [73].

Amino-bisphosphonates (N-BPs) have been used to treat bone metastases from various cancer types and osteoporosis, hypercalcemia, Paget’s disease, and other conditions for more than 40 years. In vitro and in animal cancer models, zoledronate, and alendronate, two of the most potent N-BPs, have demonstrated direct tumoricidal impact on tumor cells and immunological modulatory effects on tumor cells myeloid cells and T cells [74]. However, the systemic exposure and uses of these medicines in cancer patients are severely constrained by their quick renal clearance and sequestration in mineral bone in their free form. To overcome these pharmacokinetic obstacles, N-BPs are reformulated and enclosed in liposomal nanoparticles. In murine cancer models, liposomal zoledronate and alendronate formulations have been reported to enhance the anticancer efficacy of cytotoxic chemotherapies and adoptive T-cell immunotherapies. Here, we examine the pharmacological distinctions between liposomal N-BPs and non-liposomal N-BPs (such as clodronate), free versus liposomal N-BP formulations, and targeted versus non-targeted liposomal N-BPs, as well as the clinical and preclinical research supporting this claim. Two of the most potent N-BPs, zoledronate, and alendronate have shown direct tumoricidal impact on tumor cells in vitro and animal cancer models, as well as immunological modulatory effects on myeloid cells and T cells [75].

Here, we will discuss the functionalization techniques used with traditional liposomes to heal bone effectively (Figure 2). Liposomes have been engineered to store active targeting molecules, release medications under regulated conditions, and respond to stimuli [75,76].

Conventional liposomes typically have significant concentrations of non-bioactive lipids such as cholesterol and phospholipids, which do not naturally promote bone regeneration. Our team has created an osteoinductive liposomal formulation using oxysterols as a component of liposomes to enhance standard liposomal formulations [77]. Oxysterols are a group of steroid derivatives that have been shown to promote ossification and osteogenesis [78]. One of the most effective oxysterols for bone regeneration, 20S-hydroxycholesterol, was added to non-phospholipid liposomes made of the single-chain amphiphile, stearyl amine (SA). When applied locally to the methacrylate glycol chitosan (MeGC) hydrogel scaffold, the integration of 20S-hydroxycholesterol into SA-liposomes efficiently stimulated in vitro osteogenesis and in vivo calvarial defect healing [79].

Delivering osteogenic compounds such as purmorphamine, smoothened agonist (SAG), and signaling molecule Shh into those osteoinductive liposomes let researchers investigate their potential as a drug delivery system more recently. The increased osteogenesis and in vivo bone healing were accomplished by the simultaneous introduction of numerous osteogenic molecules [80].

### 3.1. Nanoliposomes

A new method for the encapsulation and delivery of bioactive substances is the nanoliposome, also known as the submicron bilayer lipid vesicle. There is a vast range of bioactive substances that can be added to nanoliposomes, from medications to cosmetics and nutraceuticals [81]. Nanoliposomes have potential uses in a wide range of industries, including nanotherapy (such as cancer therapy, gene transfer, and diagnosis), cosmetics, food technology, and agriculture, due to their biocompatibility, biodegradability, and nanosize. By enhancing bioactive drugs’ solubility and bioavailability, in vitro and in vivo stability, and ability to avoid unfavorable interactions with other molecules, nanoliposomes can improve the efficacy of bioactive agents [82].

Sodium bicarbonate (NaHCO_3_)-containing and tetracycline-functionalized nanoliposomes (NaHCO_3_-TNLs) have recently been designed to function as a smart “nanosacrificial layer” that can target bone surfaces and react to external secreted acidification from osteoclasts, avoiding osteoporosis. The extracellular acid–base neutralization first inhibits osteoclast function and also promotes its apoptosis, in which the apoptosis-derived extracellular vesicles containing RANK (receptor activator of nuclear factor-κB) further consummate the osteoclast apoptosis. These in vitro and in vivo results demonstrate that this nanosacrificial layer precisely inhibits the initial acidification of osteoclasts and initiates a chemical [83].

A recent development in the treatment of bone cancer is the development of novel berberine nanoliposomes with excellent encapsulation efficiency and slow-release formulations. In order to deliver slow-releasing berberine-containing nanoliposomes to bone cancer cells Saos2 (human osteosarcoma cell line), many researchers tried to find innovative ways to construct them [84]. Berberine hydrochloride is an isoquinoline alkaloids component that has been proposed for a variety of therapeutic uses. Intestinal infections, hyperlipidemia, diabetes, and arrhythmias are all treated with it. Furthermore, it prevents the development of hepatoma cell lines (H22), cervical cancer cell lines, and sarcoma cell lines in mice. However, a significant drawback is that it has poor solubility and bioavailability in the aqueous phase [85]. According to research, liposomes can be loaded with berberine to increase permeability and improve the targeting of malignant tissue. A new berberine-containing liposome with high encapsulation effectiveness and slow-release formulation in the treatment of cancer is a novel issue, even though some research used berberine-containing liposomes to deliver medicine to cells [86]. The reviewed study found that the IC50 (concentration that inhibits 50% cell growth) value of free berberine was 2.67 times higher than that of berberine-containing nanoliposomes, indicating that the proliferation of cancer cells was more inhibited by berberine-containing nanoliposomes. Additionally, because of the gradual drug release, the medication was exposed to the tumor for a longer period of time at a lower dose and with fewer injections, which increased the drug’s impact on cancer cells. Therefore, it is suggested that bone cancer cells Saos2 be treated with berberine-containing nanoliposomes [87].

Another study proposes a brand-new and improved cationic PEGylated liposomal microRNA formulation for gene delivery. Small molecule microRNA was employed in the study to test the delivery system’s capacity for encasing genes [88]. A cytotoxicity test revealed that the positive surface charge of the microRNA loaded into the liposomes in SaOs-2 cells made it more hazardous than the free form of the microRNA. Cationic liposomes could interact electrostatically with negatively charged cell membranes. The cell membrane might therefore be easily penetrated by these structures. It is crucial to improve delivery system architecture in order to boost microRNA efficiency [89]. To address this issue, several formulations incorporating polypeptides were created, described, and tested over four months for stability characteristics, size, zeta potential, and gene loading effectiveness. As a result, a high-loading microRNA-lipoplex system was created that was not agglomerated and capable of being stored at 4 °C for four months without experiencing considerable microRNA leakage. The lipoplex technology increased the transmission of microRNA into the cell while maintaining good biocompatibility with healthy cells. The PEGylated nano-liposomal formulation hence had a high potential for the systematic migration of MicroRNA and might enhance the intracellular stability of Free MicroRNA. In general, this study suggested a PEGylated nano-liposomal formulation that released slowly and was made up of mono-dispersed nanoparticles with a size in the range of 100 nm [90].

### 3.2. Tocosomes

The primary components of tocosomes, a colloidal and vesicular bioactive carrier system, are alpha tocopherols that contain phosphate groups. But they can also incorporate sterols, proteins, and polymers into their structure, just like nanoliposomes can. Alpha-tocopherol phosphate (TP), the phosphorylated version of alpha-tocopherol, is found naturally in human and animal tissues, as well as in some food ingredients [91]. The TP molecule is now known to naturally exist in several fruits, green vegetables, cereals, and dairy products, as well as various nuts and seeds [92]. A phosphate group is joined to a single hydrophobic chain (phytyl tail) that consists of three isoprene units to form TP. Two phytyl chains make up di-alpha-tocopherol phosphate (T_2_P), a TP-related compound. However, unlike phosphatidylcholine and some other phospholipids, T_2_P contains bulky isoprene side chains, which prevent the hydrophobic phytyl chains from aligning in parallel positions. As a result, the T_2_P molecule has a conical [93]. Clinical research has shown that TP and T_2_P molecules have a variety of positive effects on human health, including the ability to prevent atherosclerosis and possess cardioprotective and anti-inflammatory capabilities [94]. Additionally, the TP molecule inhibits the invasion of tumors. Additionally, studies have shown that TP can reduce the amount of lipid peroxidation products in mouse liver and plasma, as well as protect primary cortical neuronal cells from glutamate-induced cytotoxicity in vitro [95]. Along with TP and T_2_P components, tocosome formulations also contain a variety of phospholipid molecules and varying combinations of cholesterol [96,97]. They have recently been effectively used to entrap and release the anticancer medication 5-fluorouracil; therefore, they can be a great candidate for bone cancer [32].

## 4. Conclusions

Osteosarcoma is the most typical primary malignant bone tumor in children (OS). People with metastases still have a 10% to 30% chance of survival, and current treatments cannot halt the illness from progressing in these patients. It has been proposed in this context that employing nanomedicine, particularly lipid nanoparticles, could aid in treating various illnesses, including bone cancer, without being confined by existing medical procedures. Nanocarriers are advantageous as therapeutic delivery systems for anti-neoplastic medications since they can be supplied orally instead of intravenously, as in the case of lipid nanoparticles.

Due to the anatomical barrier and inadequate circulatory support in the bone, treating bone illnesses such as osteoporosis becomes very difficult. Consequently, a prospective targeted medication delivery strategy is required for its treatment. The majority of the treatments that are now on the market cannot deliver precise drugs to the desired bone location. One of the unique and efficient methods in the near future for bone-targeted delivery is the implementation of a liposome-based delivery system. This review focuses on the varied functions of delivery methods based on multifunctional liposomes in treating osteoporosis. The results of numerous in vitro and in vivo studies that used liposomes to deliver genes to the targeted bone for the treatment of osteoporosis showed that liposomes could be used as a potential non-viral vector for modulating gene expression because they are extremely effective at carrying various cargos to the targeted site. This method can be further investigated for more promising results in the treatment of osteoporosis. Additionally, liposomes can be used in a variety of ways to transport scaffolds or genes to the damaged bone in order to encourage bone regeneration. The negative consequences of medication delivery that occurs off-target are avoided by using liposomes to deliver various pharmaceuticals and targeting agents to the bone in a cell-specific and BMSCs-targeted manner. They can integrate numerous medications and distribute them through different pathways in a regulated release pattern with improved bioavailability and fewer side effects. They are also effective carriers for drug delivery systems. Overall, liposomes have shown to be a successful delivery method for treating osteoporosis, and further study in this area is warranted to confirm their positive outcomes. By considering potential structural changes, methods to get around the problems with traditional liposomes are also addressed. It will be extremely beneficial for future studies to create new-generation liposomes (such as tranfersomes, bilosomes, and exosomes) that may have an affinity for the bone. To advance this method into human clinical trials and beyond, additional research is needed because the numerous in vitro and preclinical studies now accessible in this field are still in the early phases of development. We hope for a similar kind of endeavor in the successful development of liposome-based clinical formulations for the delivery of anti-osteoporosis medications and bone-targeting treatments since liposome-based clinical formulations are already available for transporting drugs such as anticancer therapy.

## Figures and Tables

**Figure 1 biomedicines-10-03158-f001:**
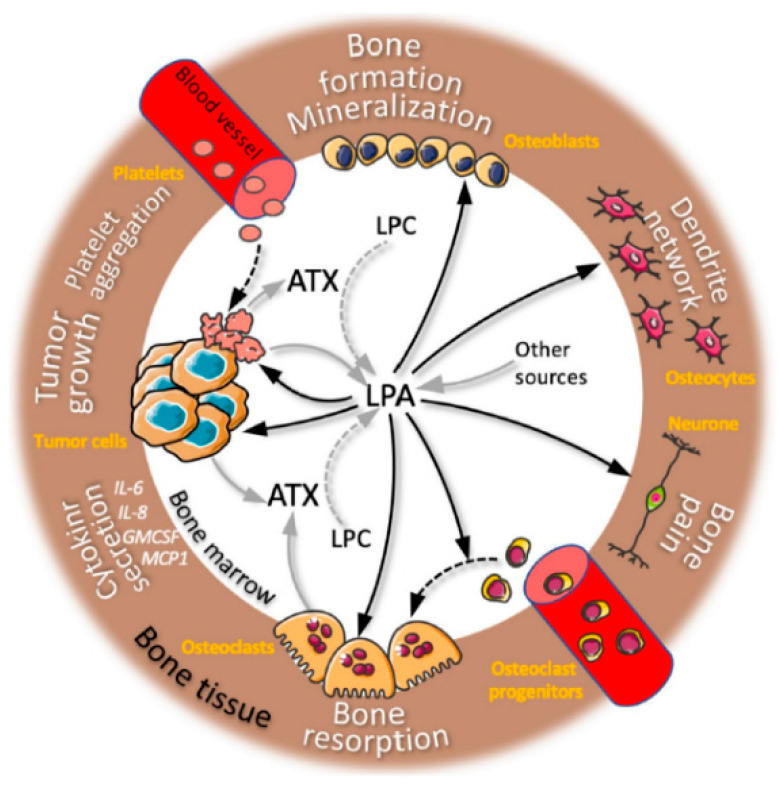
LPA activity in the microenvironment of bone metastasis: an overview of lysophosphatidic acid (LPA) activities on bone cells, cancer cells, and neurons. LPA activity on several cell types from the bone microenvironment (names in orange) results in multiple biological functions (black arrows) (text in white). Cell secretion of LPA or autotaxin (ATX) is indicated by grey arrows. Cell differentiation and platelet aggregation are indicated by dotted black arrows. ATX catabolism of LPC is indicated by dotted grey arrows. LPC: lysophosphatidylcholine [35]. Reprinted from an open access source.

**Figure 2 biomedicines-10-03158-f002:**
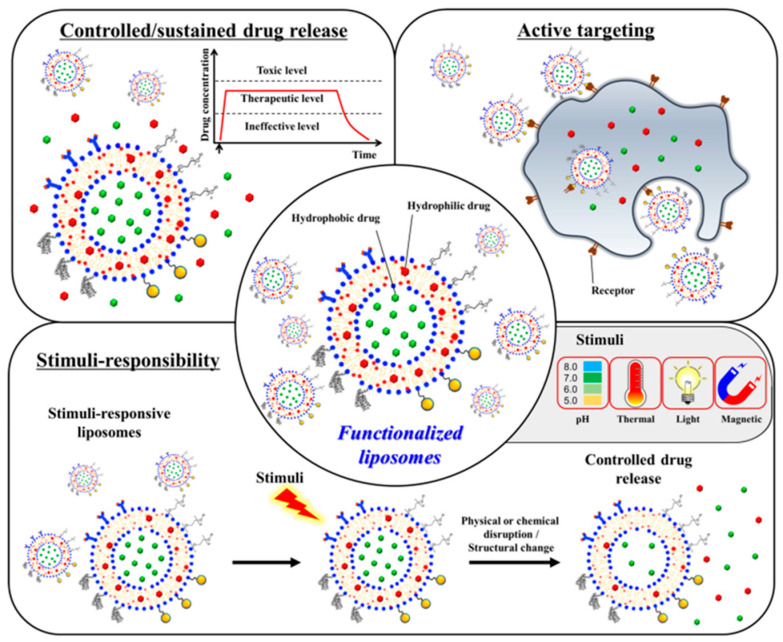
Scheme of functionalized liposomes used in bone regeneration applications. Liposomes have been designed for effective bone healing to store active targeting molecules, release medications in a regulated manner, and respond to stimuli [76]. Reprinted from an open access source.

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
