# Peer review of "Lipid Nanoparticles and Liposomes for Bone Diseases Treatment"

_biomedicines, 2022, doi:10.3390/biomedicines10123158_

Round 1

Reviewer 1 Report

Dear Authors;

Re: [Manuscript: biomedicines-2033125]

Title of Review Article: "Lipid Nanoparticles and Liposomes for Bone Diseases Treatment"

The manuscript aims to review literature on the usefulness of drug delivery systems (particularly nano-scale protocols) for the treatment of bone disease. 

Please find my comments and suggestions below:

1. The title includes liposomes in addition to lipid nanoparticles. However, the Abstract does not explain liposomes and only introduces LNP's.

2. There are several English language errors which I request authors to have a look and do required corrections. Examples include:

"... lipids nanoparticles ..." (in Abstract) remove "s" after lipid;

The last sentence in Introduction: "Considering those mentioned above, this review aims to present the lipid nanoparticles and liposomes solution for bone disease."

Line 203-205 "... and c drug loading ..." what you mean by inserting letter "c" here?

"... RES), has been the gold standard ..." needs revision and rephrasing. 

3. Among the technologies listed in the Introduction section, a novel and very recent technology is missing (i.e., Tocosome). Check literature about the new technology.

4. For the nano-adjuvants and ISCOM mentioned under section "2" consult  the recent article:   Biomedicines 2021, 9,520. https://doi.org/10.3390/ biomedicines9050520 

5. The expression: "... Lipids and surfactants, such as cholesterol, ..." (section 2, Line 74) is scientifically wrong (cholesterol belongs to the chemical group "sterols".

6. The section on liposomes  includes some mistakes e.g. liposome size range. How can we have a bilayer system smaller than 20 nanometer while the thickness of only one bilayer is about 5 nanometer?

7. Please corrects the sentence and digits and scales in the following sentence:

"... Liposomes are spherical vesicles that range in size from 0.01 to 1 m. ..."

8. Please correct the sentence about "cholesterol" in section 3 in which you wrote: "... natural phospholipids like cholesterol ..." (cholesterol is not a phospholipid. It does not have a phosphate group and glycerol and ....).

9. I suggest you include a subsection on "Nanoliposomes" and another subsection on "Tocosomes" under section 3.

10. The Conclusion is unnecessary too long. It has sections which should appear in the Introduction; e.g. "... enhanced permeability and retention (EPR) effect ..."

Best wishes

Author Response

Dear reviewer,

Thank you very much for your revisions, and I am very sorry for the late response. Down below, I present you the modifications of my review:

  1. I explained in the abstract the Liposome part of my review.
  2. I corrected the mistakes regarding the English part.
  3. A short introduction to the Tocosome was added.
  4. Some information about ISCOM was added to the suggested article.
  5. The expression was modified.
  6. The size range of the liposome was modified.
  7. The digits and scale were corrected.
  8. The cholesterol part was reformulated.
  9. I added both the suggested subsections; unfortunately, there wasn't much information about Tocosomes for bone regeneration.
  10. The conclusion was shortened.

You will find attached the revised review. Thank you very much for your help.

Reviewer 2 Report

The manuscript, entitled "Lipid Nanoparticles and Liposomes for Bone Diseases Treatment", is an overview of the use of lipid microcarriers for the treatment of bone diseases. The review is written consistently, has a sufficient number of references, and its topic is of scientific interest. It is necessary to work out the style of the English language. It is proposed to adopt the review article after minor changes.

1) Line 33. It is rather strange to call bone metastasis by the word "condition".

2) Line 45. Due to what is the reduction of toxicity in this case? Does this expression apply here?

3) A line. Perhaps, instead of the expression "Recent studies have successfully repaired ...", you should write "As a result of recent research, a group of scientists has successfully repaired..."?

4) Lines 64-67. A complex sentence, which it is desirable to reformulate, to facilitate perception.

5) Lines 100-102. It is possible that part of the phrase was not deleted by mistake during the editing process.

6) Line 165. As far as I understand, the authors apply the word "we" to other people's research. It is necessary to reformulate this.

7) Lines 265-267. The phrase "Additionally, it was non-cytotoxic to healthy osteoblast cells and promoted..." is written twice.

Author Response

Dear reviewer,

Thank you very much for your revisions, and I am very sorry for the late response. Down below, I present you the modifications of my review:

  1. The expression was changed.
  2. The sentence was reformulated.
  3. The line was reformulated according to the suggested version.
  4. The sentence was reformulated in an easy to read version.
  5. I deleted the mistaken words.
  6. I changed the noun in the text.
  7. I deleted the word and let it just once.

You will find attached the revised review. Thank you very much for your help.

Round 2

Reviewer 1 Report

Dear Authors,

Journal Biomedicines (ISSN 2227-9059) Manuscript ID biomedicines-2033125 Type Review Title Lipid Nanoparticles and Liposomes for Bone Diseases Treatment

Thanks for revising your manuscript. I believe it can be published in the current form.